# Exploring Alcohol-Related Behaviours in an Adult Sample with Anorexia Nervosa and Those in Recovery

**DOI:** 10.3390/nu16050732

**Published:** 2024-03-04

**Authors:** Zara Smalley, Maria Livanou, Bethan Dalton, Olivia Patsalos, Jessica Wilks, Johanna Louise Keeler, Janet Treasure, Ulrike Schmidt, Grace Hill, Hubertus Himmerich

**Affiliations:** 1Centre for Research in Eating and Weight Disorders (CREW), Department of Psychological Medicine, Institute of Psychiatry, Psychology and Neuroscience, King’s College London, London SE5 8AF, UK; maria.livanou@kcl.ac.uk (M.L.); bethan.dalton@kcl.ac.uk (B.D.); olivia.patsalos@kcl.ac.uk (O.P.); jessica.wilks@kcl.ac.uk (J.W.); johanna.keeler@kcl.ac.uk (J.L.K.); janet.treasure@kcl.ac.uk (J.T.); ulrike.schmidt@kcl.ac.uk (U.S.); grace.hill@kcl.ac.uk (G.H.); hubertus.himmerich@kcl.ac.uk (H.H.); 2South London and Maudsley NHS Foundation Trust, Bethlem Royal Hospital, Monks Orchard Road, Beckenham, Kent BR3 3BC, UK

**Keywords:** anorexia nervosa, recovered anorexia, alcohol use disorder, eating disorder, alcohol dependence

## Abstract

While individuals with Bulimia Nervosa (BN) and Binge Eating Disorder (BED) often present with a higher rate of Alcohol Use Disorder (AUD) than the general population, it is unclear whether this extends to AN. This cross-sectional study examined differences in alcohol-related behaviours, measured using the Alcohol Use Identification Test (AUDIT), between AN participants (*n* = 58), recovered AN (rec-AN) participants (*n* = 25), and healthy controls (*n* = 57). Statistical models controlled for age and ethnicity. The relationship between alcohol-related behaviours with ED psychopathology and with depression was also assessed. The findings indicated that acute AN participants were not at greater risk of AUD than healthy controls. However, rec-AN participants displayed greater total audit scores than those with acute AN, and more alcohol-related behaviours than healthy controls. Acute AN participants consumed significantly less alcohol than both the healthy control group and rec-AN group. No associations were found between ED psychopathology and alcohol-related behaviours in the AN group or rec-AN. This highlights alcohol as a potential coping mechanism following AN recovery. Clinicians should consider assessments for AUD and targeted interventions aimed at encouraging healthy coping mechanisms in this group. Future studies should look at alcohol use as a moderating factor for AN recovery.

## 1. Introduction

Anorexia Nervosa (AN) is a serious eating disorder (ED) with profound implications for both emotional and physical well-being. The Diagnostic and Statistical Manual of Mental Disorders (DSM-5) defines AN as a severe limitation of calorie intake leading to a significant reduction in body weight. This is often accompanied by an intense fear of gaining weight and distorted perceptions of one’s body shape, leading to body image concerns [1]. AN has a lifetime prevalence of up to 4% among females [2]. AN is linked to a diverse range of medical complications, including cardiovascular issues, electrolyte imbalances, and hormonal disturbances, all of which can become life-threatening [3]. AN is said to have the highest standardised mortality rate of all psychiatric disorders and has an unadjusted mortality rate of 5.1 deaths per 1000 person years, with 20% of these deaths attributed to suicide [4]. While the precise causes of AN remain partially unclear, research suggests that those with AN may present with an increased risk of Alcohol Use Disorder (AUD) [5,6], leading to worsened treatment outcomes and additional health concerns, including hypoglycaemia and gastrointestinal diseases [7,8].

AUD, defined by the problematic and excessive consumption of alcohol, is estimated to affect approximately 3.4% of the population across Europe [9]. The DSM-5 diagnostic criteria for AUD encompass a range of problematic patterns of alcohol consumption, such as a continuous craving to consume alcohol or an inability to manage alcohol intake [1]. Recent studies have highlighted a surge in alcohol consumption, particularly among 18–34-year-olds, during the COVID-19 pandemic, which has been linked to deteriorating mental health [10]. Co-occurring psychiatric disorders, including anxiety, major depressive disorder (MDD), and EDs, are frequently observed among individuals with AUD [5,11]. More than half of those receiving treatment for AUD present with comorbid psychiatric conditions [12]. Although, typically, the onset of MDD precedes AUD [13], studies suggest common underlying causative factors such as traits of impulsivity and genetic vulnerability [14]. However, increased alcohol intake may arise as a form of self-medication for individuals with depressive disorders [15]. For individuals with EDs, Krahn proposed that restriction of food intake could lead to changes in the reward pathways of the brain, increasing alcohol consumption through positive reinforcement [16]. Heavy alcohol consumption is associated with various physical health risks, such as infectious diseases, cancer, and cardiovascular disorders, with liver disease as the leading cause of mortality among clinical populations [17,18]. Understanding the complex interplay between AUD and co-occurring psychiatric and physical health conditions is crucial for developing effective treatment and intervention strategies.

Although it is largely undisputed that individuals with EDs present with an increased prevalence of substance use disorders (SUD) and AUD compared to the broader population [5,19], it is unclear whether this extends to AN. Studies suggest a potential link between EDs and the presentation of AUD, with purging frequency and poor psychosocial functioning serving as predictors for AUD in AN [20]. According to a longitudinal study, 10% of patients reported the onset of AUD during treatment for their ED, highlighting the risk posed by having an ED for the development of AUD. Research indicates that individuals with Bulimia Nervosa (BN) and AN binge-eating/purging (AN-BP) subtype are more prone to alcohol and substance misuse, and have a higher likelihood of developing AUD compared to healthy controls [21]. The co-occurrence of AUD and EDs may be linked to shared impulsivity traits and a loss of behavioural control proposed by addiction models [22]. Research suggests that both BN and AN-BP subtype are associated with elevated impulsivity [23,24], one of the strongest predictors for SUDs [25]. However, those with AN restricting (AN-R) subtype did not display the same traits, providing an explanation as to why individuals with AN-BP are more likely to present with AUD than those with AN-R [26]. Additionally, increased preoccupation with body weight and shape and body image disturbance increases susceptibility to the development of AUD [20].

Despite these findings, a meta-analysis involving 41 studies did not find a relationship between AUD and AN [27], contrasting the significant co-occurrence rates with AUD and other EDs, such as BN and Binge Eating Disorder (BED). However, this analysis did not account for the potential impact of age and was limited by its predominantly white female sample. Previous literature has suggested that individuals with AN-R may be less likely to develop AUD compared to their healthy counterparts [28], indicating that different clinical presentations may influence the connection between AUD and AN. In addition to impulsivity, Baker [29] noted a heightened prevalence of depression in individuals with comorbid AN and AUD when compared to AN alone. This is of significant interest given that elevated risk of depression severity has been associated with increased alcohol dependence [30].

Comorbid AUD and AN is of particular clinical relevance given the high mortality rate among individuals who experience this combination. For example, a longitudinal study indicates that alcohol use after initial assessment is the strongest and most stable predictor of death among individuals with AN [31]. This is, in part, due to the increased risk of medical complications such as osteoporosis, cardiovascular complications, and gastrointestinal diseases [7,8]. However, impaired psychosocial functioning could also exacerbate maladaptive alcohol consumption and lead to an elevated risk of suicide [31]. Findings from community samples indicate a lifetime prevalence of approximately 21% for the cooccurrence of AUD and AN [29]. This emphasises the importance of assessing individuals with EDs for comorbid AUD. A recent review suggests that the treatment of both disorders should occur simultaneously for the best outcomes [32], although it is unclear to what extent excessive alcohol consumption hinders recovery, and this should be further addressed in research and clinical practice. In addition, clinicians should be mindful of comorbid psychiatric disorders such as MDD, which has been shown to exacerbate both AUD and ED psychopathology [33,34].

This study aimed to examine the differences in alcohol-related behaviours, including dependence, alcohol-related problems, and consumption, in order to assess the overall risk of developing Alcohol Use Disorder (AUD) across three groups: (1) a group of people with acute AN, (2) those recovered from AN, and (3) healthy controls. Additionally, we studied the relationship between alcohol-related behaviours and ED psychopathology and the link between depression and alcohol-related behaviour across the three groups.

## 2. Materials and Methods

### 2.1. Study Design and Participants

A secondary analysis was performed using data from an exploratory cross-sectional study investigating inflammatory cytokines in AN, recovered AN, and healthy participants in 2018 [35]. A total of 393 potential study participants were screened. The main reasons for exclusion from the study were inflammatory or infectious diseases; living too far away; not being interested in the study; not feeling comfortable with the study; or having a normal BMI as a potential participant within the acute anorexia nervosa group. No participant had to be excluded for alcohol use disorder, even though this was an exclusion criterium.

The study recruited a total of 142 female participants aged over 18 years (age range 18–53 years). Specifically, this included a group with a primary diagnosis of AN (*n* = 59), a group who had recovered from AN (*n* = 25), and a group of healthy controls (*n* = 58). Participants were recruited using opportunistic sampling through internet advertisements, participation in previous research projects conducted by King’s College London (KCL), and Inpatient and Outpatient Eating Disorder Services at the South London and Maudsley NHS Mental Health Foundation Trust. For the AN group, participants had a primary diagnosis of AN in accordance with the DSM-5, with a body mass index (BMI) of <18.5 kg/m^2^ [1]. For the recovered group, participants were required to have previously met the diagnostic criteria for AN and to have maintained a BMI of >18.5 kg/m^2^ for the past 6 months without engaging in any disordered eating behaviours. Healthy controls had a BMI of >18.5 kg/m^2^ with no previous diagnosis of a psychiatric disorder. The exclusion criteria for all groups included being male; being under the age of 18, having an infectious, inflammatory or autoimmune disease; and being currently pregnant.

### 2.2. Procedure

Eligible participants were requested to visit the Institute of Psychiatry, Psychology and Neuroscience at KCL for a research appointment lasting up to 1 h 30 min. During the session, participants completed questionnaires to gather demographic data, such as age and ethnicity, and clinical information, including ED psychopathology, alcohol-related behaviours, and depression. Participants’ height and weight were measured via a non-invasive Inbody S10 machine, Biospace Co., Ltd., Des Moines, IA, USA. It is worth noting that blood samples were collected for the original study [35] but were not incorporated into the current study.

Informed consent was obtained from all participants prior to data collection. The original study was approved by the Ethics Committee of London City and East (Reference: 17/LO/2017; date 12 February 2018), in line with the Declaration of Helsinki.

### 2.3. Measures

#### 2.3.1. Demographic Information

Demographic information was collected, including age and ethnicity by self-report. Participants’ current height and weight were measured to calculate BMI.

#### 2.3.2. Eating Disorder Examination-Questionnaire

The Eating Disorder Examination-Questionnaire (EDE-Q; [36]) is a 28-item self-report measure used to assess the attitudes and behaviours of people with Eds over the last 28 days. Questions measuring behaviour frequency were rated on a 7-point Likert scale from 0 (no days) to 6 (every day). Participants rated the severity for ED psychopathology on a 7-point scale, reflecting the distress associated with the condition. The EDE-Q subscales are Restraint, Eating Concern, Weight Concern, and Shape Concern. Subscale scores were calculated by adding the ratings together and dividing them up by the sum of the items within each subscale. Global scores were derived from the sum of all subscale scores divided by the number of subscales. Higher scores reflected greater ED psychopathology and scores of 4 or higher indicated participants were within a clinical range. The EDE-Q has shown strong concurrent validity with other measures and good test–retest reliability [37]. The internal consistency of our sample was good (Cronbach’s α = 0.97).

#### 2.3.3. Beck Depression Inventory

The Beck Depression Inventory (BDI-II) [38] is a self-report questionnaire which was employed to assess the severity of depressive symptoms. It consists of 21 items that measure the cognitive, emotional, and physical symptoms associated with depressive disorders. Each item presents 4 statements to describe symptom severity over the past two weeks, with assigned values from 0–3. Higher scores indicate greater severity of depression, with overall scores indicating little to no depression (0–13), mild mood disturbance (14–19), moderate depression (20–28), or severe depression (29–63). The BDI-II has been shown to have good test–retest reliability and good concurrent validity among multiple populations [39]. The internal consistency of our sample was good (Cronbach’s α = 0.96).

#### 2.3.4. Alcohol Use Disorders Identification Test

The Alcohol Use Disorders Identification Test (AUDIT) is a self-report measure consisting of 10 items [40]. It was administered to assess frequency and quantity of alcohol use, alcohol dependence, and alcohol-related problems in our sample. The scoring for each question ranges from 0 to 4, with 0 indicating ‘never’ and 4 indicating the highest frequency, for example, ‘daily’. As such, higher scores indicate a greater likelihood of harmful alcohol consumption. Items 1–3 assess the regularity and quantity of alcohol consumption and items 4–6 measure alcohol dependence. Lastly, items 7–10 assess alcohol-related problems by focusing on the adverse effects of alcohol consumption, such as feeling guilty or causing injury. The AUDIT has showed good test–retest reliability [41]. The internal consistency of our sample was good (Cronbach’s α = 0.81).

### 2.4. Statistical Analysis

Two participants were excluded from data analysis due to incomplete questionnaire data. Therefore, data from a total of 140 participants were included in the analyses.

All statistical analyses were conducted using SPSS version 29. The normality of the data was assessed using the Shapiro–Wilk test. For normal data, ANOVA tests were used for the comparisons of demographic data between the three groups. For skewed data, the Kruskal–Wallis test was employed to compare demographics among groups. The AUDIT scores were negatively skewed, so a Quade non-parametric analysis of covariance was employed to assess the differences in alcohol-related behaviours across the three groups, with age and ethnicity as covariates given the significant differences in ethnicity composition in the sample. Hypothesis testing was conducted using a significance level (α) of 0.05, indicating a threshold for statistical significance. A two-tailed test was employed, and *p*-values less than 0.05 were considered significant. If the total model was significant, post hoc Dunn’s pairwise tests were carried out for the three groups. We opted for Quade’s non-parametric analysis instead of ANCOVA or regression due to the observed skewness in the data and the violation of homoscedasticity assumptions. Quade’s test is robust at handling non-normal and heteroscedastic data, providing a more reliable analysis under these specific conditions. 

To explore the relationship between alcohol use and ED psychopathology, correlations between the AUDIT questionnaire and EDE-Q, and AUDIT and BDI-II were analysed using the Spearman rank correlation coefficient (rho).

## 3. Results

### 3.1. Participant Characteristics

Table 1 summarises the demographic characteristics of participants with AN, recovered AN and healthy controls, and group comparisons. 

Groups were similar in terms of age, although ethnicity significantly varied across the three groups. The healthy group had significantly more participants from an ethnic minority group when compared to acute AN and recovered AN, who were predominantly white.

As expected, acute AN participants had a significantly lower BMI and higher ED psychopathology than participants with recovered AN (*p* < 0.001) and healthy controls (*p* < 0.001). However, individuals recovered from AN were similar to healthy controls in terms of BMI and eating disorder psychopathology. Acute AN participants also had higher depressive symtomotology than those recovered from AN (*p* < 0.001). Recovered AN participants had significantly worse depressive symtomotology than healthy controls (*p* = 0.025). 

### 3.2. Differences in Alcohol-Related Behaviours between AN, Recovered AN, and Healthy Controls

Table 2 displays the descriptive and inferential statistics of Quade’s ANCOVA models for AUDIT variables. 

#### 3.2.1. Total AUDIT Score

As demonstrated in Table 2, there were significant differences in the total AUDIT score between the three groups. Post hoc pairwise tests indicate that those recovered from AN were at greater risk of AUD compared to acute AN. However, there were no differences in the total AUDIT between acute AN and healthy controls and recovered AN and healthy controls. 

#### 3.2.2. Alcohol Consumption

There were significant differences in alcohol consumption between the three groups. Post hoc pairwise tests indicate that those recovered from AN consumed significantly more alcohol than those with acute AN, and healthy controls consumed significantly more alcohol than those with acute AN. However, there were no significant differences in the level of alcohol consumed between healthy controls and recovered AN.

#### 3.2.3. Alcohol Dependence

There were no significant differences in the level of alcohol dependence between groups. 

#### 3.2.4. Alcohol-Related Problems

There were significant differences in the level of alcohol-related problems between groups. Post hoc pairwise tests indicated that participants recovered from AN reported significantly more alcohol-related problems compared to healthy controls. However, there were no significant differences found between the acute AN and healthy control group and the acute AN and recovered AN group. 

### 3.3. Assessing the Relationships between EDE-Q and BDI, EDE-Q and AUDIT Scores

Table 3 presents the correlations of the AUDIT scores and the EDE-Q subscales across the three groups, and the AUDIT scores and the BDI-II.

No significant correlations were found between AUDIT and EDE-Q, or AUDIT and BDI-II scores within the AN group or the recovered AN group. However, in the healthy control group, alcohol dependence was positively associated with EDE-Q global and shape concern subscale scores. Alcohol-related problems were positively associated with EDE-Q global and shape concern sub scale scores, as well as BDI-II scores. 

## 4. Discussion

### 4.1. Summary and Significance of the Main Results

This cross-sectional study sought to examine the differences in alcohol-related behaviour in a sample of individuals with AN, recovered AN, and healthy controls using the AUDIT screening tool. Relationships between alcohol-related behaviours, eating psychopathology, and depression, were investigated across the three groups.

Our findings indicate that there were no significant differences in AUDIT scores between the acute AN group and healthy control group and the recovered AN group and the healthy control group. Additionally, there were no significant differences in alcohol dependence between the three groups. However, those recovered from AN were at greater risk of AUD, as shown by their significantly greater total AUDIT scores compared to the scores of those with acute AN, and experienced greater alcohol-related problems compared to healthy controls. Alcohol-related problems might include distress, memory impairment, and alcohol-related injuries [40]. Those with current AN consumed significantly less alcohol than those who were recovered and healthy controls. However, it is possible that the high calorie content of alcohol is an incentive to not to drink in those that are currently unwell. The current findings contradict a singular study that suggests individuals with EDs were less likely to be diagnosed with SUD and MDD twenty-two years after recovery [42]. However, authors did not distinguish between AN and BN and their sample was limited to females seeking treatment in the 1980s. Previous studies have highlighted that depressive, anxious, and obsessive compulsive symptomatology is present following recovery from AN [43], but there is a notable gap in the current literature regarding the relationship between alcohol use and recovery from AN. This raises an important question as to whether alcohol is used as a coping mechanism in this group and may be related to residual ED psychopathology. This is especially relevant given that the recovered AN group had significantly more depressive symptoms than healthy controls in the current study. 

However, ED psychopathology and AUDIT scores were not linearly related in our correlation matrix. 

Consistent with a previous meta-analysis that did not find an association between AUD and AN [27], the present study found that current AN participants were not at an increased risk of AUD when compared to healthy controls. However, this is contrary to some existing literature that suggests an increased risk of substance misuse and AUD among individuals with AN [8,19]. A substantial body of literature suggests individuals with AN-BP subtype are at significantly greater risk of AUD than those with AN-R subtype [5,20]. As such, differing findings may be attributed to the composition of our sample, which predominantly consisted of individuals with AN-R subtype (86%). However, it is interesting to note that differences were observed in the recovered AN group, despite all individuals having a history of AN-R. This raises intriguing questions about the potential long-term effects and variations within this specific subgroup.

Although there were observed differences between the three groups, it is important to note that all groups had mean total AUDIT scores that are considered low risk as per the thresholds [40]. This suggests that, while there are notable variations in alcohol-related problems and behaviours across the groups, the overall risk for problematic alcohol use may be relatively low in the studied population.

Correlational analyses revealed no significant relationships between ED psychopathology and alcohol-related behaviours in the acute AN group and recovered AN group. This may point to differing underlying psychological mechanisms for maladaptive alcohol behaviours, such as shared biological risk factors, depressive symptomatology, or underlying traits of obsessive compulsiveness [14,43]. This warrants further investigation with a larger and more diverse sample, particularly in males and across multiple ethnicities.

Healthy controls displayed positive correlations between ED psychopathology, depression, and alcohol-related behaviours. Specifically, increased shape concern and EDE-Q global score were associated with an increase in alcohol dependence and alcohol-related problems. This is supportive of previous literature that suggests poor body image is a risk factor for excessive drinking in non-clinical samples [44]. In addition, the use of binge drinking as a coping mechanism seems to be more common in women with increased shape concern who are at risk of developing EDs, although failing to meet the full diagnostic criteria [45]. Dietary restraint has also been associated with binge drinking among women [46], increasing the likelihood of adverse consequences of alcohol use.

The relationship between depression and AUD has been extensively discussed in the literature, with either disorder doubling the risk of developing the other [47]. Supporting our results, positive associations between depressive severity and total AUDIT score have previously been found in a sample of healthy individuals [48].

In the supplementary analysis focusing on exclusively on Caucasian participants (see the Appendix A for more details) notable differences emerged in the AUDIT scores between the three groups, mirroring the findings of the primary analysis. Additionally, individuals with acute AN exhibited significantly lower total AUDIT scores than healthy controls. Furthermore, individuals with AN also reported significantly fewer alcohol-related problems when compared to those with recovered AN. These findings align with existing literature suggesting that individuals with anorexia nervosa may be less likely to suffer from alcohol use disorder than healthy controls [28], which is, again, potentially attributable to concerns related to calorie consumption and the avoidance of alcohol due to its caloric content. Additionally, these results may indicate differing attitudes towards alcohol use in individuals with AN across ethnicities. This highlights the need for further exploration of cultural influences on alcohol consumption and attitudes in the context of eating disorders, given that they are relatively understudied in this population.

### 4.2. Implications

The present study has several implications. Firstly, clinicians should be aware of the potential for maladaptive alcohol use following AN recovery and alcohol misuse should be considered a possible comorbid feature during this time. Integrating alcohol-related assessments into follow-up protocols to identify and address potential problems may be of value. This proactive stance would facilitate the development of targeted intervention strategies to address concerns related to alcohol use. Psychoeducation, along with targeted interventions aimed at promoting healthy coping mechanisms, may be beneficial for those with a history of both AN and BN, given the existing body of research [20,49]. This is of great importance considering the increased risk of physical health complications associated with AUD [9].

Significant positive correlations in the healthy group highlight the importance of screening for signs of EDs and other psychological factors in individuals seeking treatment for AUD.

### 4.3. Strengths and Limitations

It is important to note that the present study had several limitations. Firstly, minoritised ethnic groups were over-represented in the healthy control group. Specifically, 35.7% of the participants were of Asian heritage. It is essential to acknowledge this demographic composition as a potential confounding factor, given that individuals of Asian heritage exhibit gene variants in alcohol dehydrogenase, affecting alcohol tolerance and therefore, potentially, alcohol use [50]. However, by controlling for ethnicity in the primary analysis, this concern was partially mitigated. Furthermore, the lack of diversity in the AN sample, primarily consisting of white females, limits the generalisability of the findings to males and other ethnicities. Given the heightened prevalence of AUD amongst males [51] and underrepresentation of males in ED literature, the inclusion of males should be a focus for future studies. Similarly, the different subtypes of AN were not adequately represented, and as such we were not able to determine the relationship between alcohol-related behaviours and AN in those with AN-BP. 

Despite the limited sample size, the present study provides new insights regarding the association between alcohol-related behaviours and recovery from AN. As such, a key strength of the study is the inclusion of individuals who have recovered from AN who are understudied in this clinical population. 

## 5. Conclusions

While no significant differences were found between acute AN and healthy controls, recovered AN individuals showed greater alcohol-related problems compared to healthy controls and higher total AUDIT scores compared to acute AN. These findings warrant further investigation to understand the underlying mechanisms behind the relationship between recovered AN and AUD. Future studies should recruit a larger sample size to corroborate these findings and should aim for greater sample representation in terms of ethnicity and AN subtype. This will allow for greater statistical power and robust analysis to reveal potential risk factors associated with different subtypes. Longitudinal analysis may provide deeper insight into the temporal relationship between AN and AUD, including those who have recovered. This could enable the identification of factors contributing to the possible association between AUD and AN following recovery. Future research should be complemented with qualitative findings to gain a deeper understanding of the attitudes towards alcohol use in those who have recovered from AN. It could be useful to compare those with recovered AN to individuals with BN to determine whether the observed relationships are specific to AN or more broadly applicable. This is of particular relevance given that BN and AUD have been consistently associated with one another [26,49].

## Figures and Tables

**Table 1 nutrients-16-00732-t001:** Demographic and clinical characteristics of AN, recovered AN, and healthy participants.

Demographic Characteristic	Anorexia (*n* = 58)	Recovered Anorexia (*n* = 25)	Healthy Controls (*n* = 57)	Group Comparisons
Age (years (M, SD))	26.50 (7.99)	26.40 (6.08)	23.61 (3.99)	*F* (2) = 0.895 *p* = 0.413
Ethnicity (%)				
White	89.7	92	52.6	*χ*^2^ (2) = 35.705 *p* < 0.001 *
Asian	3.4	4	35.1	
Black	0	4	3.5	
Mixed	6.9	0	3.5	
Other	0	0	5.3	
AN Subtype (%) Restrict Binge/Purge				
	84.5 15.5	100 0	- -	- -
BMI (M, SD)	15.95 (1.26)	20.64 (2.00)	21.42 (2.09)	*F* (2) = 149.805 *p* < 0.001 *
EDE-Q Global (M, SD)	3.90 (1.19)	0.98 (0.83)	0.62 (0.74)	*H* (2) = 94.094 *p* < 0.001 *
BDI-II (M, SD)	30.67 (11.26)	11.24 (9.47)	4.57 (5.35)	*H* (2) = 89.602 *p* < 0.001 *

Note: Percentages are based on the total number of participants in each group. AN = anorexia nervosa; BDI-II = Beck Depression Inventory; BMI = Body Mass Index; EDE-Q = Eating Disorder Examination-Questionnaire; M = mean; SD = standard deviation. * indicates significance at the 0.05 level.

**Table 2 nutrients-16-00732-t002:** Descriptive statistics of AUDIT scores with the statistical significance of the group comparison from the ANCOVA model’s three groups: AN, recovered AN, and healthy controls.

Variables	Anorexia Nervosa	Recovered AN	Healthy Controls	Group Comparisons			
	M, SD	M, SD	M, SD	Total Model	AN vs. HC	AN vs. Rec-AN	HC vs. Rec-AN
Alcohol Consumption	2.99 (2.93)	4.36 (2.69)	3.56 (2.27)	*F* (2) = 4.040 *p* = 0.020 *	*p* = 0.022 *	*p* = 0.017 *	*p* = 0.541
Alcohol Dependence	0.47 (1.79)	0.64 (1.08)	0.26 (0.58)	*F* (2) = 2.726 *p* = 0.069	*p* = 0.433	*p* = 0.021	*p* = 0.089
Alcohol-Related Problems	1.40 (2.87)	1.56 (1.87)	0.49 (0.80)	*F* (2) = 5.058 *p* = 0.024 *	*p* = 0.175	*p* = 0.090	*p* = 0.007 *
AUDIT Total	4.84 (6.35)	6.56 (4.72)	4.32 (3.16)	*F* (2) = 3.255 *p* = 0.040 *	*p* = 0.168	*p* = 0.013 *	*p* = 0.152

Note: This table illustrates the descriptive statistics and group comparisons for the scores of the Alcohol Use Identification Test (AUDIT) across three groups in our sample. M and SD represent mean and standard deviation, respectively. * indicates significance at the 0.05 level.

**Table 3 nutrients-16-00732-t003:** Correlations between AUDIT scores and EDE-Q and BDI-II scores across AN, recovered AN, and healthy participants.

			Variables			
Variables	EDE-Q Restraint	EDE-Q Eating Concern	EDE-Q Shape Concern	EDE-Q Weight Concern	EDE-Q Global	BDI-II
**Anorexia (*n* = 58)**						
Alcohol Consumption	−0.052 [−0.313, 0.217]	0.073 [−0.196, 0.332]	0.029 [−0.238, 0.293]	−0.113 [−0.368, 0.157]	0.007 [−0.259, 0.272]	−0.107 [−0.362, 0.163]
Alcohol Dependence	0.124 [−0.147, 0.377]	0.123 [−0.148, 0.376]	0.174 [−0.096, 0.421]	0.103 [−0.167, 0.359]	0.151 [−0.119, 0.401]	0.039 [−0.229, 0.302]
Alcohol-Related Problems	−0.087 [−0.345, 0.183]	0.007 [−0.259, 0.272]	−0.006 [−0.271, 0.260]	−0.046 [−0.308, 0.223]	−0.014 [−0.278, 0.253]	−0.049 [−0.310, 0.220]
AUDIT Total	−0.015 [−0.279, 0.252]	0.090 [−0.180, 0.347]	0.051 [−0.218, 0.312]	−0.078 [−0.337, 0.191]	0.040 [−0.228, 0.302]	−0.088 [−0.346, 0.181]
**Recovered Anorexia (*n* = 25)**						
Alcohol Consumption	−0.038 [−0.437, 0.373]	−0.072 [−0.464, 0.343]	−0.064 [−0.457, 0.351]	−0.063 [−0.457, 0.351]	−0.090 [−0.478, 0.327]	0.000 [−0.406, 0.405]
Alcohol Dependence	0.076 [−0.340, 0.467]	0.000 [−0.405, 0.406]	−0.81 [−0.471, 0.335]	0.006 [−0.401, 0.410]	−0.045 [−0.443, 0.367]	0.116 [−0.304, 0.498]
Alcohol-Related Problems	0.064 [−0.350, 0.458]	0.020 [−0.388, 0.422]	−0.065 [−0.458, 0.350]	−0.001 [−0.406, 0.405]	0.000 [−0.406, 0.406]	0.098 [−0.320, 0.484]
AUDIT Total	−0.005 [−0.410, 0.401]	−0.083 [−0.472, 0.334]	−0.087 [−0.476, 0.330]	−0.103 [−0.488, 0.316]	−0.108 [−0.492, 0.311]	0.019 [−0.390, 0.421]
**Healthy Controls (*n* = 57)**						
Alcohol Consumption	−0.114 [−0.370, 0.159]	−0.167 [−0.416, 0.106]	−0.010 [−0.277, 0.259]	−0.122 [−0.378, 0.151]	−0.103 [−0.361, 0.170]	0.075 [−0.200, 0.338]
Alcohol Dependence	0.243 [−0.026, 0.480]	0.182 [−0.900, 0.429]	0.265 * [−0.003, 0.498]	0.182 [−0.090, 0.429]	0.276 * [0.220, 0.648]	0.217 [−0.057, 0.460]
Alcohol-Related Problems	0.253 [−0.016, 0.488]	0.155 [−0.118, 0.406]	0.333 * [0.071, 0.551]	0.126 [−0.147, 0.381]	0.278 * [0.011, 0.508]	0.332 * [0.068, 0.553]
AUDIT Total	−0.044 [−0.308, 0.227]	−0.072 [−0.334, 0.200]	0.085 [−0.187, 0.345]	−0.060 [−0.323, 0.211]	0.009 [−0.276, 0.260]	0.172 [−0.103, 0.423]

Note: Values in brackets indicate the 95% confidence interval for each correlation. Emboldened values and * indicate correlation is significant at the 0.05 level (2-tailed).

## Data Availability

The datasets analysed during the present study are available from the corresponding author upon reasonable request. The data are not publicly available because the participants were not asked to consent to the present article before inclusion into the study.

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
