# Peer review of "Exploring Alcohol-Related Behaviours in an Adult Sample with Anorexia Nervosa and Those in Recovery"

_nutrients, 2024, doi:10.3390/nu16050732_

Round 1

Reviewer 1 Report

Comments and Suggestions for Authors

Some minor revisions are suggested:

-       In the abstract it is stated that individuals with ED often present with higher rates of AUD as compared to the general population and that it is not clear whether this applies to patients with an AN diagnosis. Perhaps it would be useful to also include the specific diagnostic categories in which higher AUD rates have been found, instead of referring to ED in general.

-       It is not clear whether participants’ current weight and height were only self-reported or if they have also been assessed by a physician. If they were only self-reported, this should be included as a limitation of the study, as patients might under-report or over-report their weight. 

-       In the statistical analyses section, the p-value chosen for the analyses should be specified.

-       Table 1 not only includes demographic characteristics of patients but also some clinical data related to the mean scores on EDE-Q and BDI-II, therefore the title of the table (“demographic characteristics of AN, recovered AN and healthy participants”) should also include this information (i.e, “demographic and clinical characteristics of AN, recovered AN and healthy participants”).

-       It would be useful for the reader to highlight in bold the significant values in table 1 (as it has been done for tables 2 and 3).

Reviewer 2 Report

Comments and Suggestions for Authors

The manuscript of Smalley et al. presented the results of an on-site cross-sectional study that investigated the possible correlation between anorexia nervosa and alcohol use disorder in patients, in a similar pattern found between eating disorders and alcohol use disorder. Generally, data analysis has been appropriately conducted. However, concerns remain about the unbalanced demographic characteristic distribution amongst experiment groups, which may require additional statistics to strengthen the claims.

Specifically, there is a credible concern over the ethnicity (White/Asian) ratio in the control group. As noted in the group comparison of Table 1, the P value from the chi-square test is highly significant. A quick calculation using the 2x2 contingency table of ANvsctrl and RANvsctrl also had the same results. By some chance, I had some experience in practising clinical studies in the Greater London area in East and South Asian communities. As a result, I better understand the culture and subsequent divergent attitudes on body shape and alcohol use between Asian and European communities, and I consider such cultural covariants could have strong influences at the behavioural level. Nevertheless, considering that the population of healthy controls is relatively large (57), random sub-sampling of the Asian individuals in the control group to make the ethnicity proportion comparable (again, chi-square test) to the other two groups should provide an efficient and statistically sound method, and it should not be difficult to re-calculate the associated tables based on the amended values used.

Minor points:

1. The use of italics is inconsistent in the tables and table legends. Essentially, all math symbols (e.g., n, F) should be in italics;

2. The formatting of float numbers (numbers with decimals) is inconsistent in the tables. In Tables 1 and 2, the floats are presented with the prefix 0; however, in Table 3, the floats are presented without the prefix 0.

Round 2

Reviewer 2 Report

Comments and Suggestions for Authors

With the addition of the SI and the paragraph in the discussion section, the quality of the manuscript has improved sufficiently to warrant publication.